# Lithium Metal under Static and Dynamic Mechanical Loading

**Ed Darnbrough \*** and **David E. J. Armstrong**

Department of Materials, University of Oxford, Parks Road, Oxford OX1 3PH, UK
*   Correspondence: ed.darnbrough@materials.ox.ac.uk

**Abstract:** Macro-scale mechanical testing and finite element analysis of lithium metal in compression have been shown to suggest methods and parameters for producing thin lithium anodes. Consideration of engineering and geometrically corrected stress experiments shows that the increasing contact area dominates the stress increase observed during the compression, not strain hardening, of lithium. Under static loading, the lithium metal stress relaxes, which means there is a speed of deformation (engineering strainrate limit of $6.4 \times 10^{-5} \, \text{s}^{-1}$) where there is no increase in stress during compression. Constant displacement tests show that stress relaxation depends on the initial applied stress and the amount of athermal plastic work within the material. The finite element analysis shows that barrelling during compression and the requirement for high applied stresses to compress lithium with a small height-to-width ratio are friction and geometric effects, respectively. The outcomes of this work are discussed in relation to the diminishing returns of stack pressure, the difficulty in closing voids, and potential methods for designing and producing sub-micron lithium anodes.

**Keywords:** lithium metal; mechanical properties; thin metal anodes





## 1. Introduction

The mechanical behaviour of lithium is important for solid-state battery operation and the manufacture of thin metal anodes [1,2]. Thin metal anodes with no excess lithium are needed to realise solid-state batteries made of lithium metal, offering a high theoretical capacity (3860 mAh/g and ∼900 Wh/L). This has the potential to improve the safety and range of batteries for electric vehicles [2–4]. The creation of anodes thinner than 10 μm through mechanical processing requires a full understanding of the mechanical properties of lithium metal and how it is effected by different modes of deformation.

Early work on lithium mechanics focused on its response to stack pressure, which has been shown to be positively related to the critical stripping current; however, recent work has shown that this can cause a failure in electrolytes during plating [5–7]. More extensive research into lithium has shown that the material's response to mechanical deformation is dependent on the dislocation density and strain rate [8–11]. The literature suggests that the dislocation density required for lithium to deform plastically under tension is lower than that needed for compression or nanoindentation [10,12]. The authors also observed increased strength on small scales as a result of a low number of dislocations available to induce plasticity [11,13,14]. Experimental studies on the compression of lithium observed counter-intuitive results, where smaller creep rates were observed under larger loads [9,15]. This, combined with the observation that lithium with a small height-to-width ratio increased yield strength [10,16], means that producing thin film anodes requires an understanding of how lithium deforms under compressive loads and how that knowledge can be used to optimise anode production.

In this paper, we investigate the behaviour of lithium metal under compression and link those findings to how a thin anode would behave under stack pressure (in operando) and during formation (in preparation).

## 2. Methods

All of the sample preparation and testing were conducted within an Ar glovebox atmosphere with $\leq$0.1 ppm of $H_2O$ or $O_2$. Tests were conducted on lithium metal (purity of 99.9% purchased from Sigma Aldrich, St. Louis, MO, USA) and for each sample ~0.1 g was formed into a square sectioned block with an aspect ratio of 2:1 using a parallel sided vice. The samples were tested in a microtest set-up with a load gauge with a maximum of 1 kN produced by DEBEN UK Ltd., Bury St. Edmunds, UK. A minimum of two samples were used for each type of test described below. The evolution of stress in lithium metal was studied using constant displacement rate compression and static displacement (stress relaxation) tests. We defined yield as the departure from elastic deformation and flow as a stress capable of producing continued increase in strain with no increase in stress.

### 2.1. Compressive Forging and Strain Rate Observations

Deformation tests were recorded with engineering stress and strain (using the displacement recorded by the DEBEN system and initial sample dimensions measured by Vernier callipers). Four tests were conducted per sample with displacement rates of 0.1, 0.2, 0.3, and 0.5 mm/min, which equated to strain rates of approximately 0.11, 0.22, 0.33, and $0.55 \times 10^{-3}$ s$^{-1}$, respectively.

### 2.2. Stress Relaxation Observations

For the stress relaxation tests, samples were repeatedly loaded (10 times at a displacement rate of 0.1 mm/min) to a chosen engineering stress (0.4 MPa or 0.6 MPa) via compression, and then the displacement was held constant while the change in load was recorded to investigate the stress relaxation.

### 2.3. In-Situ Observation of Compression

Geometric stress and strain tests were conducted using the displacement and geometry extracted from a video of the test using a MATLAB 2023a routine (shown in Appendix A.3) from files recorded with a Thor Labs Scientific Camera (Kiralux 1.3 MP CMOS Compact Scientific Camera, Newton, NJ, USA) with a MVL7000 18–108 mm EFL, f/2.5, for 2/3″ C-Mount lens. Plastic flow tests were conducted at a displacement rate of 0.2 or 0.5 mm/min, and stress relaxation loading was conducted at 0.1 mm/min.

### 2.4. Finite Element Analysis

Finite element analysis was conducted in Abaqus/CAE 2016 software to create a model consisting of two steel platens between which a lithium sample was compressed up to 50% strain. The model scale was chosen to produce distances in mm and stresses in MPa. The material properties provided to the steel and lithium were as follows: elastic moduli ($E$) 200 GPa, poisons ratio ($\nu$) 0.3; and $E$ 7.8 GPa, $\nu$ 0.28, respectively [10,17,18]. Lithium was provided a linear plastic behaviour with a yield of 1 MPa with no strain hardening as observed experimentally in this work. A range of friction coefficient between the steel and lithium elements were used (0.01, 0.1, 0.5, and 0.9) to simulate the differences from a perfectly lubricated interface to a dry high friction metal–metal interaction. For all models, a C3D8R Element type was used.

## 3. Results

### 3.1. Compressive Forging and Strain Rate Observations

The constant displacement rate tests displayed an increase in engineering stress that was dependent on the speed of the compression (Figure 1a,b). Compressive engineering stress and strain were calculated without taking any change in sample cross sectional dimensions into account. This was unlikely to fully describe the behaviour of the material under an applied stack pressure in a battery application. The response of the material illustrated two distinct regions: (1) a near instantaneous increase in stress that indicated an elastic response, and (2) a slower increase in stress with time that indicated a plastic

response. The stress rate (gradients green in Figure 1a) increased with the increasing strain rate (gradients in green Figure 1b) in the elastic region as expected for a metal, as the elastic modulus was fairly constant [8]. The constant displacement rate also led to an increasing engineering stress in the plastic regime (due to an increase in contact area and/or work hardening of the material) and the rate of that stress increase was linearly related to the engineering strain rate (Figure 1c). The gradient of the linear relationship between the two rates ($dd\sigma/dd\epsilon$) was ~2 MPa.

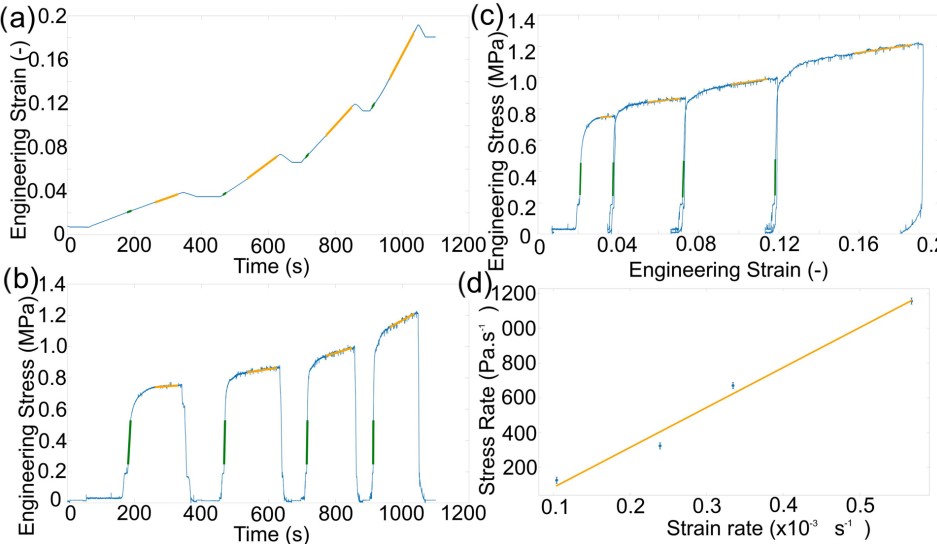

**Figure 1.** (**a**) Engineering stress vs. time during compression with four different strain rates with linear fitting of the elastic regions (green) and where the material behaves plastically (orange). (**b**) Engineering strain vs. time during compression with a linear fit of the same regions relating to the plastic deformation. (**c**) Engineering stress vs. strain plot. (**d**) The resultant plot of the engineering stress rate vs. engineering strain rate during plastic deformation, where standard error in strain rate is shown by the marker size and the error in stress rate is provided by vertical bars.

This relationship suggests that there was an engineering strain rate for plastic deformation at which the engineering stress would not increase ($6.4 \times 10^{-5}$ s$^{-1}$), requiring no additional load to continue plastic formation. At speeds below this, the measured engineering stress decreased with time. This, in turn, means that when the material was held at a constant strain, the load was reduced (known as stress relaxation), which will be discussed further in the next section. This material response implied a slow deformation could be used to change the geometry of lithium metal for use as thin anodes, where the work hardening and increase in contact area were counteracted by stress relaxation within the material.

### 3.2. Stress Relaxation Observations

Figure 2a shows tests where lithium was loaded and then held at a given displacement. The lithium displayed a stress relaxation response (Figure 2b) that was similar to the logarithmic decay observed in other metals (above 0.6 of their melting temperature), which was attributed to the elastic energy in the system being converted into plastic work, leading to a reduction in stress that tended towards a plateau at a value related to the amount of athermal deformation in the material [19]. Figure 2c shows the stress–strain curve, highlighting that during stress relaxation, there was no change in the strain, resulting in a vertical drop in the graph. Engineering stress relaxation curves with time did not fit perfectly with the logarithmic decay, as it did not account for any prior change in contact area during loading, but could be used to extract a suitable measure of athermal stress. Figure 2d shows the increase in athermal stress with repeated 100 s stress relaxations extracted from the decay curves.

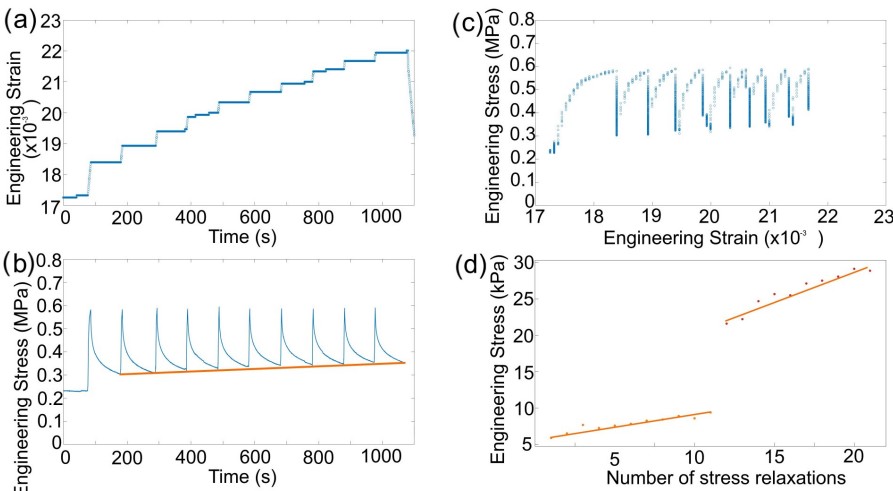

**Figure 2.** (**a**) A typical set of ten 100 s constant strain tests showing elastic loading and stress relaxation. The orange line connects the stress value after 100 s of each test as a guide for the reader. (**b**) Engineering stress vs. strain of the stress relaxations and (**c**) the step loading and strain hold segments. (**d**) Increase in athermal stress after successive 100 s stress relaxations, first set with an initial stress of 0.3 MPa and the second with an initial stress of 0.6 MPa. The orange line represent the linear fit of each data set.

The speed at which athermal defects are created in lithium during a 100 s stress relaxation test depends on the magnitude of the initial loading. With an initial engineering stress of 0.3 MPa, the athermal increased at a rate of 40 Pa·s$^{-1}$, but this increased to 60 Pa·s$^{-1}$ when the initial load for relaxation was 0.6 MPa (Figure 2c). This explains the result from Masias et al. [9], where higher constant stress tests resulted in lower strain rates after a given time because the effective contact area had a greater increase and more athermal stress was accumulated in the same time period at higher loads. This means that the mechanical history of a lithium sample affects the amount of relaxation because of the amount of plastic work/athermal stress present. This also suggests that any metallic battery anode with an assumed stack pressure (engineering stress) applied at the beginning of cycling is unlikely to be able to retain that stress over time.

*3.3. In Situ Observation of Compression*

Measuring the changing contact area and barrelling allows the true material response to compression to be understood. The geometrically corrected stress with constant displacement rates displayed a flow stress that was dependent on the strain rate (Figure 3). The compressive strain initially caused a linear elastic response, then a departure from that linear response (plastic yield), and work hardening, which led to an increase in stress until a point where the stress was no longer increasing (flow stress). Figure 3 shows that increasing the strain rate increased the flow stress. The data collected here (0.75 MPa for $\sim$3 $\times$ 10$^{-4}$ and 1 MPa for $\sim$8 $\times$ 10$^{-4}$ ) fit well with the compression data reported by Masias et al. [10].

The flow stress observed in lithium at lower displacement rates indicates there was a dynamic equilibrium between work hardening and relaxation, so when the speed increased, a new higher yield stress was observed (0.75 MPa at $\sim$3 $\times$ 10$^{-4}$ increasing to 1 MPa at $\sim$6 $\times$ 10$^{-4}$) due to an increase in dislocation density before a flow stress was reached for the higher speed [8,12]. This strain rate sensitivity was observed previously, but there has been little discussion of the recovery process [9,20]. Conducting repeated stress relaxation tests after the material has been flowing indicates a new yield point, as shown at $\sim$910 s in Figure 3a. These observations suggests that the linear increase in engineering stress observed in the constant displacement rate tests (Figure 1) was dominated by the increase in the effective area and that the recovery mechanism counteracted the work hardening

under load when the dislocation density reached an equilibrium dependent on the speed. This agreed with Fan et al. for the dislocation density evolution.

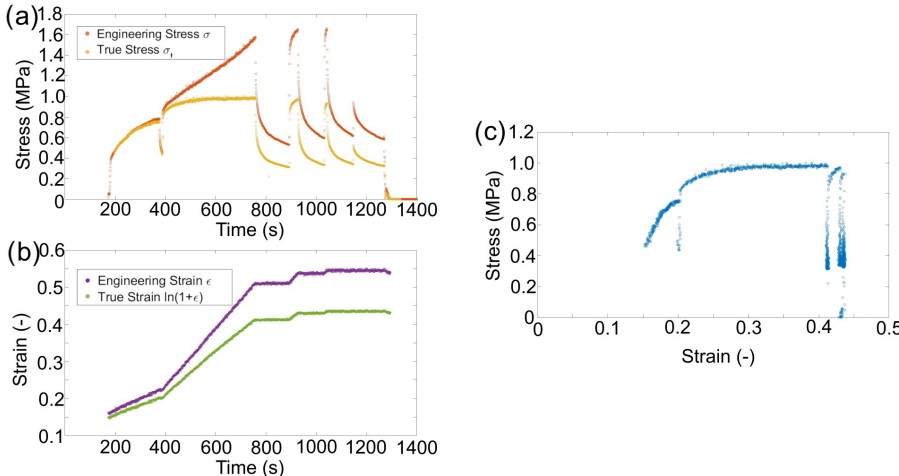

**Figure 3.** (**a**) Compressive strain of a sample with time. (**b**) Geometrically corrected stress evolution with different strain rates showing that a flow state occurs and the real stress relaxation. An initial loading of 0.2 mm/min (200–400 s) equating to a strain rate of $\sim 3 \times 10^{-4}$ followed by an increased displacement rate of 0.5 mm/min equating to $\sim 6 \times 10^{-4}$ ($\geq 400$ s). (**c**) The true stress plotted against true strain.

The amount of relaxation in 100 s, shown in Figure 3a, was similar ($\Delta\sigma = 0.64, 0.64$, and 0.59 MPa for the first three holds) and loading to half the initial stress caused a significantly smaller stress reduction in 100 s ($\Delta\sigma = 0.29$ MPa), but reached a similar absolute value of 0.33 MPa that was related to the athermal stress in the material (Figure 3b).

This indicates that the scale of the recovered stress when the strain rate was 0 (stress relaxation tests) was dependent on the load at the beginning of the relaxation and tended to a value that was dependent on the athermal stress (Figure 4). Different loadings were shown to reach the same stress after 100 s if the athermal stress in the material was the same (Figure 4a). However, the same loading with different amounts of athermal stress produced different gradients and end points in the 100 s relaxation tests (Figure 4b). The athermal stress could be increased by repeated loading (S.I. Figure A3b,c show true stress data that mirrors that shown above in the engineering stress tests) and leaving the samples under no load appears to effect the yield stress of the material more than the athermal stress, as shown by elastic loading and plastic loading after 12 h in Ar atmosphere at 25 °C Figure 4c,d. Long stress relaxations do increase the athermal stress but less than repeated loading, S.I. Figure A5. The reduction in yield agrees with room temperature annealing reducing the dislocation density [8]. Considering the athermal stress before and after stress relaxation in a number of different circumstances highlights that each loading and relaxation provided a moderate change in athermal stress, but there were two significant outliers highlighted in green and blue (Figure 4e). The largest increase came after a period of plastic loading (green) and the largest decrease came from the sample being left unloaded (blue) (Figure 4f). Comparing the increase in athermal stress with the peak stress during the active compression of lithium for $\sim 500$ s shows a positive relationship, as expected (S.I. Figure A4). If we assume this was a linear relationship with time and load (more long term experiments are needed to suggest otherwise), then the increase in athermal stress over 500 s was approximately 16% of the peak loading value.

This suggests that the work hardening produced in the active deformation of the lithium material to increase the yield was as a result of dislocation pinning, which could be overcome thermally (at room temperature that equated to $\sim 0.6$ of the melting temp), but less of the athermal defects formed by elastic to plastic relaxation could be undone at room temperatures.

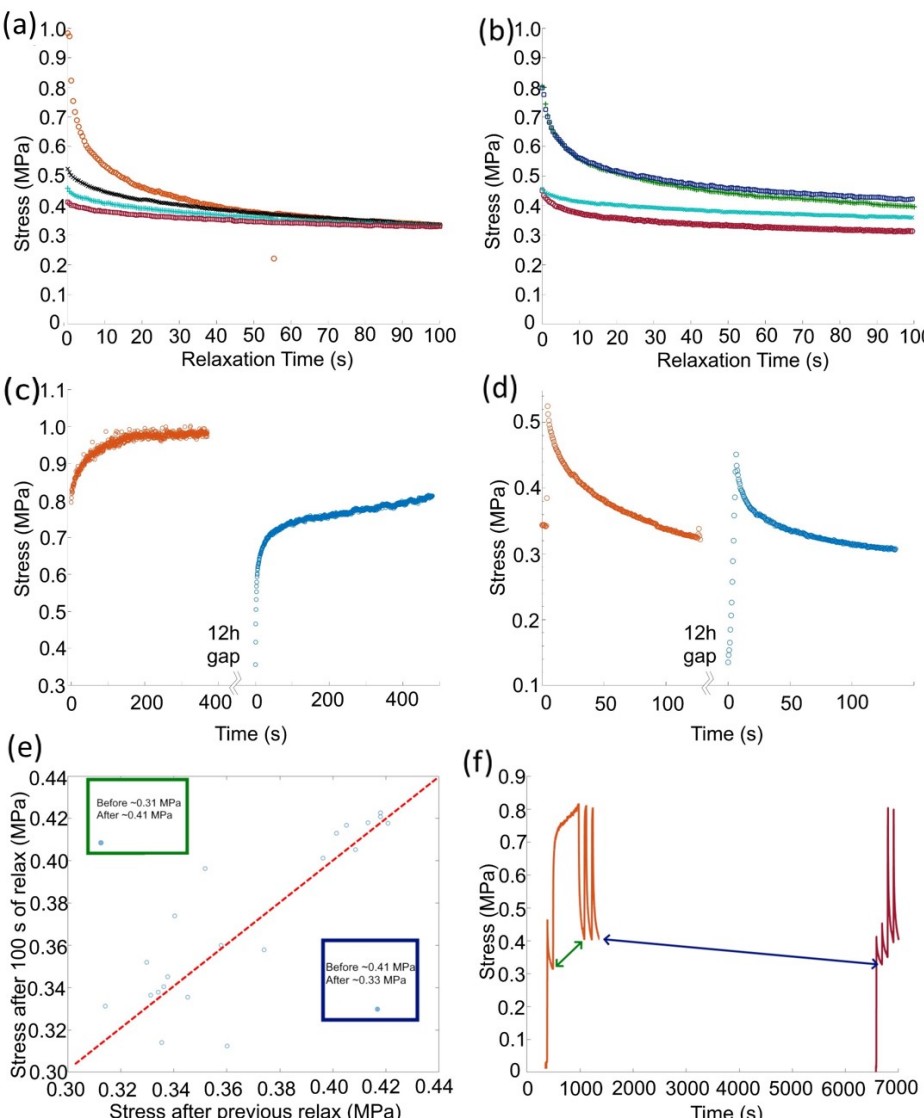

**Figure 4.** Geometrically corrected stress relaxation examples from samples with different plastic histories illustrating: (**a**) four different loading stresses can produce the same value after 100 s of stress relaxation if the athermal stress in the material was the same, (**b**) two examples of the same loading stress producing a different final stresses after 100 s due to different amounts of athermal stress in the sample. (**c,d**) The plastic response and stress relaxation before (orange) and after (blue), respectively, the sample is left at room temperature for 12 h under no load. (**c**) The yield point of the material reduces (blue) compared with an earlier yield (orange), and (**d**) the stress relaxation is similar after 12 h of no load with only a small reduction in the athermal stress. (**e**) The athermal stress measured before a relaxation test vs. the stress after 100 s of relaxation, the dashed line shows a 1:1 ratio of stresses. (**f**) Highlights the time periods and the before and after stresses that lead to the two outliers shown in (**e**).

During stress relaxation, there was no observed change in the sample geometry, but the plastic deformation prior could be followed by in situ observation of the macro compression tests. Previous papers have touched on the 'barrelling' effect, whereby the contact area did not grow, but the waist of the sample increased in size [10,21]. This was due to the frictional forces occurring at the surface between the lithium and steel (or other patten material) being used to apply the pressure. The change in geometry caused the distribution of stress to become more complicated and it has been shown that this frictional effect could cause the strength of the material to be overestimated [22].

As the test progressed after barrelling, the lateral forces caused by the friction became greater than the yield strength of the material and it started to forge [23]. Forging of a material under otherwise uniaxial compression is the indication that the stress and strain experienced by a region in a sample varies, depending on the position and size of the sample, with the central region experiencing the largest stress and the outer regions being lower. Critical to the size of the stresses involved was the ratio of the sample height-to-width, as the shorter the deformed piece in compression, the greater the internal stress gradients must achieve.

The finite element analysis (FEA) of lithium metal compressed between two steel platens displayed barrelling and the effect of the friction coefficient ($\mu$) between the steel and lithium (Figure 5). The overlayed FEA element edges (Figure 5a) illustrates that when the friction coefficient was 0.01, the sample deformed with near parallel sides (blue) and there was a minimal increase in barrelling with a friction coefficient greater than 0.5 (yellow and red). The $\mu$ of 0.01 provided a uniform radius increase of 1.04 mm, but the higher $\mu$ of 0.5 provided a range from a minimum at the interface of 0.60 mm to a maximum of 1.17 mm in the centre. The minimal difference between the geometry produced with a $\mu$ of 0.5 and 0.9 suggests that in this range, the frictional stress at the interface exceeded the yield stress of the material, producing the same result. This was supported by the uniform Von Mises stress at all points in the sample with a value of 1 MPa, and both samples displaying a similar distribution of hydrostatic stress ($1/3(\sigma_1 + \sigma_2 + \sigma_3)$) and shear stresses within the $\mu$ 0.5 and 0.9 samples (Figure 5b,c). The distribution of stresses illustrates that where the friction was low ($\mu = 0.01$), the stresses were uniform throughout the module and where the increased friction led to 'barrelling', there was a broader range of stress present in the sample. The maximal shear stresses that contributed to plasticity were on the surfaces of the sample and the maximal pressure was in the core of the sample (Figures A8 and A9). The higher internal pressures indicate that a larger applied load would be required to produce the 50% reduction in sample size for $\mu$ 0.1 vs. 0.5, but no increase was required between a $\mu$ of 0.5 and 0.9. When this was compared with the experimental result, there was minimal 'barrelling' observed during compression (Figure 5d), suggesting that the friction coefficient in our tests was less than 0.5 and there was no strengthening artefact introduced to our results as a result of friction [22].

Using FEA to interrogate the size effect, we can see when $\mu$ was 0.5 and the sample height-to-width ratio decreased by a factor of 5, the 50% compression led to an increase in radius of 0.81 to 1.26 mm (Figure A10). This was the same size barrelling effect, but with a greater absolute final radius and occurred with a factor of five increase in maximal internal pressure that was then located centrally in the sample, not at the surface, as with the taller samples. When the sample height was only 100 µm, the maximum internal pressure for a 50% reduction in height reached 135 MPa. This suggests that the strengthening effect observed when the samples were shorter ws less related to the friction and yielding of a material at the interface, but was determined by the internal pressure that maybe related to the amount of plastic work being done on the material. This strengthening size effect during compression has been observed in other work [10,16,24]. Stallard et al. attributed this increase in strength at low sample heights to the classical 'fiction hill' observed in forging, where there is a peak in both the lateral and axial stresses in the centre of the sample. Further discussions of this distribution and how friction can increase the required applied pressure for forging are contained within the Appendix A.1.

The FEA visualisation of the regions under the greatest shear and normal stress during deformation dependent on $\mu$ show that minimising friction could reduce the applied pressure required to reduce the sample height by 50%. However, as shown by Masias et al., using mineral oil as a lubricant, there was still a size effect to overcome at low dimensions [10]. This means, for metal anode battery materials where the aim is to form low height-to-width ratio parts, forging would require pressures greatly exceeding the yield stress of bulk lithium as with all metals.

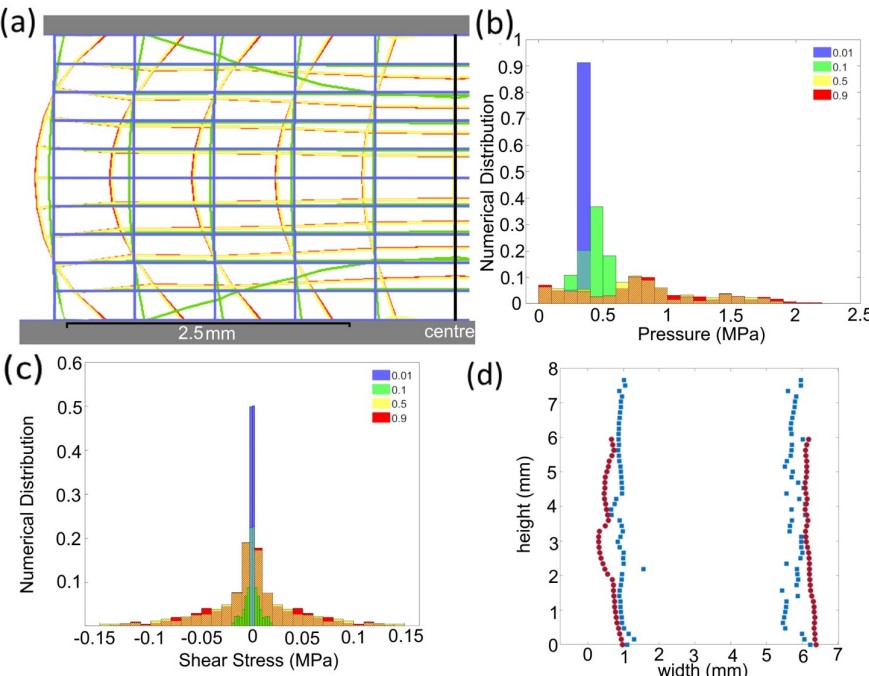

**Figure 5.** (**a**) The extracted FEA element edges (from Figure A11) for samples under 50% compression with different $\mu$ values (blue 0.01, green 0.1, yellow 0.5, and red 0.9), (**b**) the distribution of hydrostatic stress magnitudes from FEA nodes, and (**c**) the distribution of shear stresses from FEA nodes. In the histograms (**b**,**c**), regions where bars overlap are denoted with a striped fill. The $\mu$ values for (**a**–**c**) are 0.01 blue; 0.1 green; 0.5 yellow; and 0.9 red. (**d**) shows minimal barrelling in the edge profile of a sample before testing (blue) and after a 20% compression (red) as extracted from images using Appendix A.3, scale 64 pixel per μm.

Although this suggests that forging is a sub-optimal method for the initial production macro-scale micron-thick lithium anodes, this work is still of paramount importance when considering the closing of voids at the interface during cell manufacture (formation pressures) and cycling (applied stack pressures).

## 4. Discussion

### 4.1. Stack Pressure Effect on Thin Lithium Anodes

This work has shown that stress relaxation is significant in lithium metal, such that an anode, even after high pressure forming, will not retain a stack pressure as applied by static deformation. Even retaining stresses (below yield) to enhance creep during cycling would require dynamic compression. A deformed spring could provide this initially; however, as the anode deforms, the stress held in a spring would reduce and, in time, there would effectively be minimal stack pressure. Increasing the athermal stress could be used to mitigate this stress relaxation, but will increase the proportion of pinned dislocations and reduce the amount of dislocation creep in the material. Additionally, the increased strength of the material at small scales suggests that the closing of voids with moderate stack pressures is unlikely. Void growth was suggested to be mitigated by higher stack pressures bringing more lithium to the interface than what is leaving [5,25]; however, recent observations have shown that lower stack pressures result in a greater cycling performance than higher ones by avoiding failures in the solid electrolyte [7,26]. Work by Zhang et al. suggested that any stack pressure will be intensified locally with rough anode–electrolyte interfaces that can lead to localised contact, producing smaller effective contact areas than expected [27,28]. Krauskopf et al. suggested that a formation pressure of 400 MPa will produce a perfect electrode–electrolyte interface with minimal resistance [25]. The requirement for a pressure exceeding the yield strength of the material by three orders of magnitude suggests that to close the smallest voids requires stress significantly above

the bulk yield stress. This may be due to a combination of effects: the small scale of the initial contact provides enhanced strength due to the size of the lithium being deformed; then, as 'flattening' progresses, the material being forged to close the voids now contains significant plastic work, increasing the strength; finally, the effective height-to-width ratio of the material being forged geometrically enhances the strength of the material.

### 4.2. Producing Thin Lithium Anodes

Physical vapour deposition techniques have been shown to produce good quality thin metal anodes, but this is not a viable route for mass manufacture [29]. Casting thin metal anodes to create a desired geometry is challenging due to the difficulty in working with liquid lithium and the fact is wets surfaces poorly [30,31]. A dominant method for forming lithium for use in batteries is rolling; however, that has a similar exponential relationship to compression when attempting to create very thin samples (Appendix A.2). In this case, friction is required for the foil not to be rejected from the process, and tension can be applied to reduce the average pressure. Rolling must be performed in multiple passes as the percentage reduction on each pass is limited, which, with the requirement for high pressure, leads to high production costs for thin lithium foils [32]. Rolling can technically produce thin lithium anodes, but this has the side effect of modifying the microstructure, as shown by LePage et al. [20], that in turn can dominate the mechanical response at a small scale [8]. The higher yield and reduced strain hardening displayed by the rolling direction vs. the transverse direction suggest an anisotropic dislocation arrangement in the material due to rolling, which could lead to non-uniform cycling behaviour across the anode. Annealing and further mechanical deformation could be used to modify the microstructure after rolling. The rolling will produce a preferential [100] grain texture normal to the foil surface [33] that will dominate the elastic response and result in an effective elastic modulus around 5 GPa in the stack pressure axis [8,18]. A combination of this work and previous has shown that the plastic response will be dominated by the stored dislocation density and the proportion of those that contribute to athermal stress [8]. The variance in yield strength reported in the literature suggests that tensile deformation produces less dislocation density in lithium than compression, and our work has shown that annealing (even at room temperature) can be used to reduce the mobile dislocation density. In this study, we have shown that stress relaxation is so prevalent that it must be accounted for when considering how lithium metal will act as an anode. The athermal stress (the level the material relaxes to) can be increased by long stress relaxations, repeat loading, and extended plastic flow. A combination of rolling, tension, compression, and annealing could therefore be used to tune the response of the anode to stack pressure with high mobile dislocation density aiding creep for higher stripping currents or a high immobile dislocation density to reduce stress relaxation and retain stack pressure for longer operation.

## 5. Conclusions

This work highlights the lithium mechanical behaviour and how that could impact the production and cycling of thin metal anodes. Stack pressure is not capable of forming perfect interfaces, but some work has suggested it is enough to compensate for void growth during moderate stripping currents when the temperature is increased, to improve diffusion kinetics and creep properties [34]. Further to this, our findings suggest that the stress relaxation in lithium metal would mean any applied stack pressure is unlikely to be present after a week of creating a solid state cell without active loading. This suggests forming high quality interfaces and small scale anodes requires prior manufacture.

If forming thin lithium anodes by compression, the best strain rate is $6.4 \times 10^{-5} \, s^{-1}$, which will cause plastic flow of the lithium without an increase in the required load. Our works suggests that dynamically loading lithium to hold it at a geometrically corrected stress of 1 MPa for 8 h would increase the athermal stress to avoid stress relaxation, which would otherwise reduce the stack pressure after cell construction.

Forging small height samples requires increasing pressure due to the geometry of forging materials with a low height-to-width ratio. FEA has been used to indicate the regions that will have the highest stress while under compressive force, which means that during cycling an anode may produce different creep behaviour based on location, as stress distribution is non-uniform.

We have discussed that a selection of compressive forging, tensile, and rolling mechanical loading can produce different material characteristics that could be optimised for cycling performance.

**Author Contributions:** Conceptualization, E.D. and D.E.J.A.; Methodology, E.D.; Software code, E.D.; Validation, E.D.; Formal Analysis, E.D.; Investigation, E.D.; Data Curation, E.D.; Writing: Original Draft Preparation, E.D.; Writing: Review & Editing, E.D. and D.E.J.A.; Visualizations, E.D.; Supervision, D.E.J.A.; Project Administration, E.D. and D.E.J.A.; Funding Acquisition, D.E.J.A. All authors have read and agreed to the published version of the manuscript.

**Funding:** This research was funded in whole, or in part, by the UKRI Faraday Institution [SOLBAT FIRG026 and FIRG056].

**Data Availability Statement:** The raw and processed data required to reproduce these findings cannot be shared at this time due to technical and time limitations. The data can be shared through direct contact with the corresponding author.

**Acknowledgments:** This research was funded in whole, or in part, by the UKRI Faraday Institution [SOLBAT FIRG026 and FIRG056]. For the purpose of Open Access, the author has applied a CC BY public copyright licence to any Author Accepted Manuscript version arising from this submission. The authors acknowledge use of characterisation facilities within the David Cockayne Centre for Electron Microscopy, Department of Materials, University of Oxford, alongside financial support provided by the Henry Royce Institute (Grant ref EP/R010145/1).

**Conflicts of Interest:** The authors declare no conflict of interest.

## Appendix A

*Appendix A.1. Effect of Sample Geometry on Forging*

Assuming a finite frictional force the pressure ($P$) within a compressed material (of dimension $h \times b \times L$) is dependent on the position within the material [23]. Considering an element of material with the compressed sample allows a relationship between the pressure due to an external load, and the distance from the centre ($x$) of the forged material can be given as follows:

$$P(x) = -2k exp(\frac{2\mu x}{h})$$

where $k$ is the shear yield stress, $\mu$ is the friction coefficient, h is the height of the sample, and $x$ is the position within in the width ($b$) of the material. This pressure profile provides a maximum in the centre and a minimum at the edges as the material slides at the interface, with the pattern further expanding the contact area. The total force applied to a compressed material is equal to the integral of the pressure over the total width that can then be related to the average pressure ($P_{av} = F/bL$).

$$\frac{P_{av}}{2k} = -\alpha(exp(\frac{1}{\alpha} - 1)); \alpha = \frac{h}{b\mu}$$

The relationship between the average pressure (stack pressure) and the height-to-width ratio can be plotted with different friction coefficients to show that the compression of samples requires a stack pressure that exceeds the shear yield of the material.

The frictional forces ($P(x)\mu$) at the interface can only increase to a maximum of $2k$ where the material starts to shear in favour of sliding. This is called the stick–slip transition and is located within the sample such that the edges (where pressure is lower) slide and

the material interface at the central region is stuck. In the "stick" region the pressure is given by the following:

$$P(x) = -2k(1 + \frac{x}{h})$$

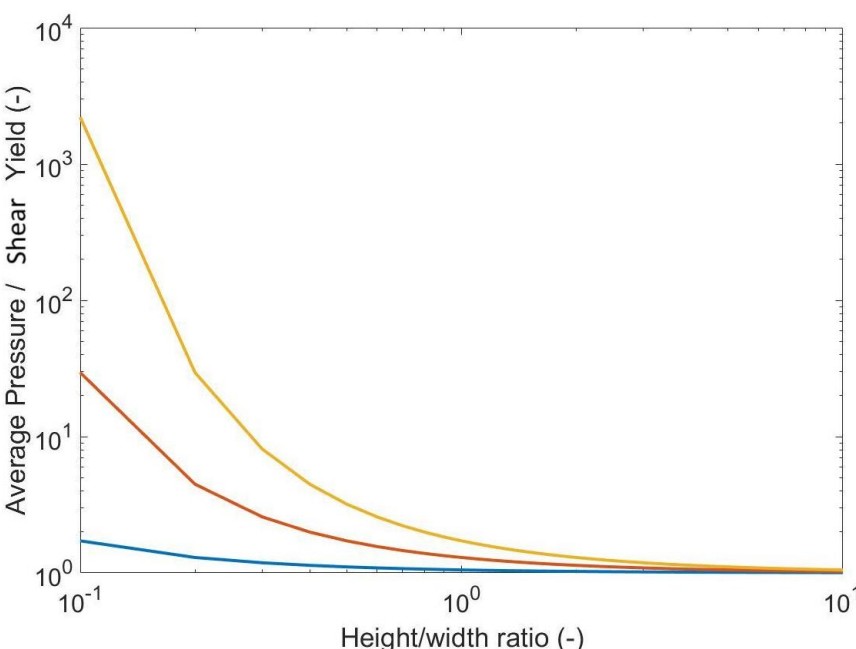

**Figure A1.** Relationship between the average pressure and yield stress with a constant coefficient of friction with varying height to width ratio. Friction coefficients: 0.1 blue, 0.5 orange, 0.9 yellow.

*Appendix A.2. Pressure during Rolling*

Rolling a sample between two rollers produces plastic deformation with a similar 'friction hill' to forging, resulting in an average pressure over the distance between the first and final contact ($L$) of:

$$\frac{P_{av}}{2k} = -\alpha. * (exp(\frac{1}{\alpha} - 1)); \alpha = \frac{\bar{h}}{L\mu}$$

where $\bar{h} = \frac{h_{in} + h_{out}}{2}$.

As the technique requires friction to pull the sample through, there is a limit to the reduction possible in a single pass, given by the following:

$$\Delta h = \mu^2 R$$

where $R$ is the radius of the roller used, which also increases the distance $L$ and thus the average pressure required to roll the material. The average pressure can be reduced by placing the sample being rolled under tension, because the total stress on the material will equal the plastic yield:

$$|P| + |\sigma| = 2k$$

$$P_{av} = -\alpha. * (exp(\frac{1}{\alpha} - 1)(2k - \frac{\sigma_{front} + \sigma back}{2})); \alpha = \frac{\bar{h}}{L\mu}$$

*Appendix A.3. MATLAB Geometry Extraction*

The files for the MATLAB routine used to extract sample geometry and displacement during tests from videos can be found here: https://github.com/EdDarnbrough/DEBEN-and-ThorCam (accessed on 20 December 2023).

These files import the video file (.avi created by the ThorLabs ThorCam software version 3.6.0.6) and the force data file (.cvs created by the DEBEN Microtest software version 6.3.40) using their modification dates to align them in real time. Then, an image taken pre-test and the first frame of the recording are used to define the real size of the pixels in the video. The image pre-test is taken zoomed out to capture the sample and the full DEBEN platens in view. The exact size of the platens is known and there is no change in the sample geometry between the pre-image and the first frame, allowing a scaling factor to be applied to the video images that are recorded zoomed in enough to maximise the resolution for observing changes, without the need for a scale present throughout. The experimental set-up is created with the camera looking at one face of the sample with a background that is much brighter than the platens and the sample, which then allows the edges to be found by fitting an error function to the pixel intensity.

$$ecf_{fit} = \frac{1}{2}range(pixels(x)erf\left(\frac{x-i}{\sqrt{\delta}}\right) + min(pixels(x))) + \frac{range(pixels(x))}{2}$$

where $pixels(x)$ is the function describing the intensity in a column or row from the video frame, $i$ is the centre point of the error function, and $\delta$ is a value that determines how abruptly the error function changes from the minimum to the maximum value. The best fit is determined by finding the value of $i$ with the minimum sum of residuals. The error function fit is used with optimised region selection, and the direction is dependent on the edge being measured within an image.

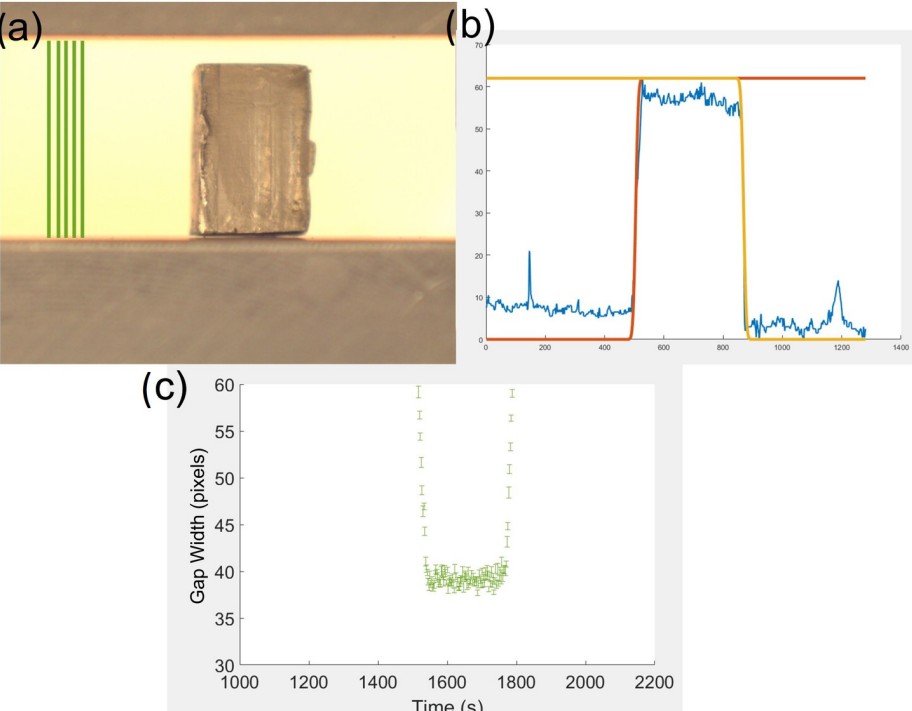

**Figure A2.** (**a**) Example image taken from the beginning of a test video recording with the pixels used for platen gap measure marked by green lines. (**b**) A plot of the arbitrary intensity extracted from a vertical line of pixels (blue) with a forward (orange) and backward (yellow) error function fitted. (**c**) Shows the extracted gap between platens throughout a compression test from the positions indicated in green in Figure (**a**) that are averaged to provide the confidence in the position measurement, shown as error bars.

The deformation applied by the platens is monitored by measuring the movement of the upper and lower platen separately when considering the sum of the 50 columns of

pixels on the left edge of the image, which contain no sample throughout the entire test (Figure A2).

The sample geometry is considered once vertically and horizontally to extract the height and width of the sample, respectively. It is assumed here that the sample responds uniformly to deformation and so any change in observed width will be mirrored in a change in the depth of the sample. The sample is selected to be in focus, which can lead to a defocus on the platen edge; however, the value recovered for the central position of the edge appears unaffected by this, as shown by the insensitivity of the measured value with varying the value of $\delta$ from 10 to 1000. For all of the results reported here, a value of 600 was used for both the platen and sample edges, as it provides the smallest minimum sum of residuals when conducting the sensitivity test.

*Appendix A.4. Further Data on Material Tested*

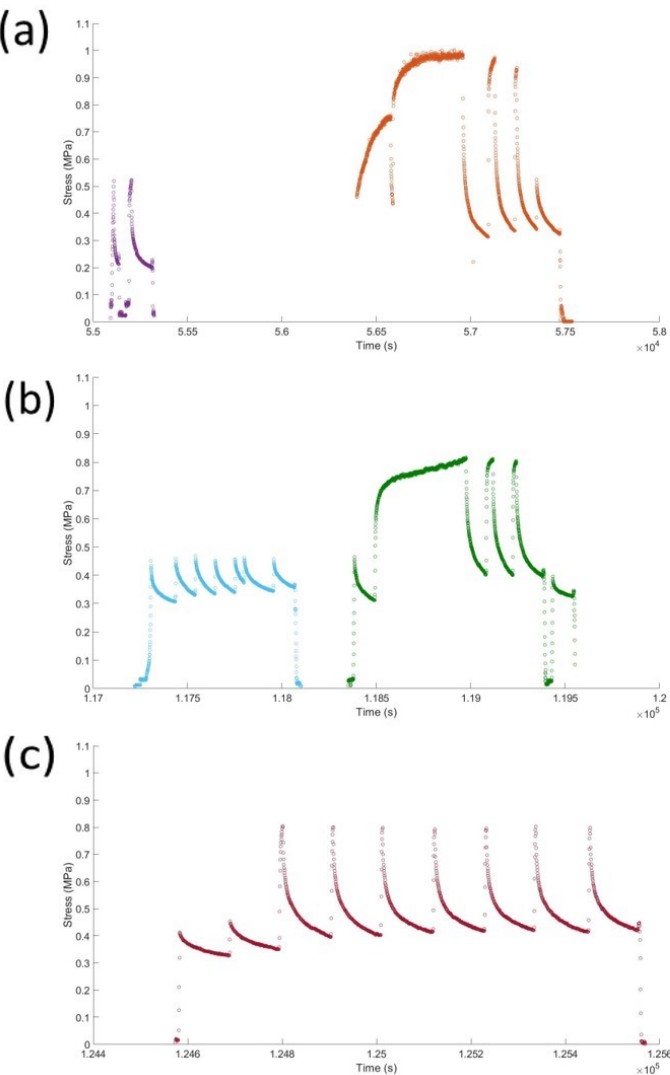

**Figure A3.** The geometrically corrected stress of a sample tested in the afternoon (**a**), the next morning (**b**), and the next afternoon (**c**). The time shown in seconds is converted from a standard 24 h clock, so the reader can relate the tests to one another. This shows the response of the material approximately 18 h after initial testing and the evolution of the stress relaxation response. Different colours are used to differentiate different tests, between these times the sample is under no load.

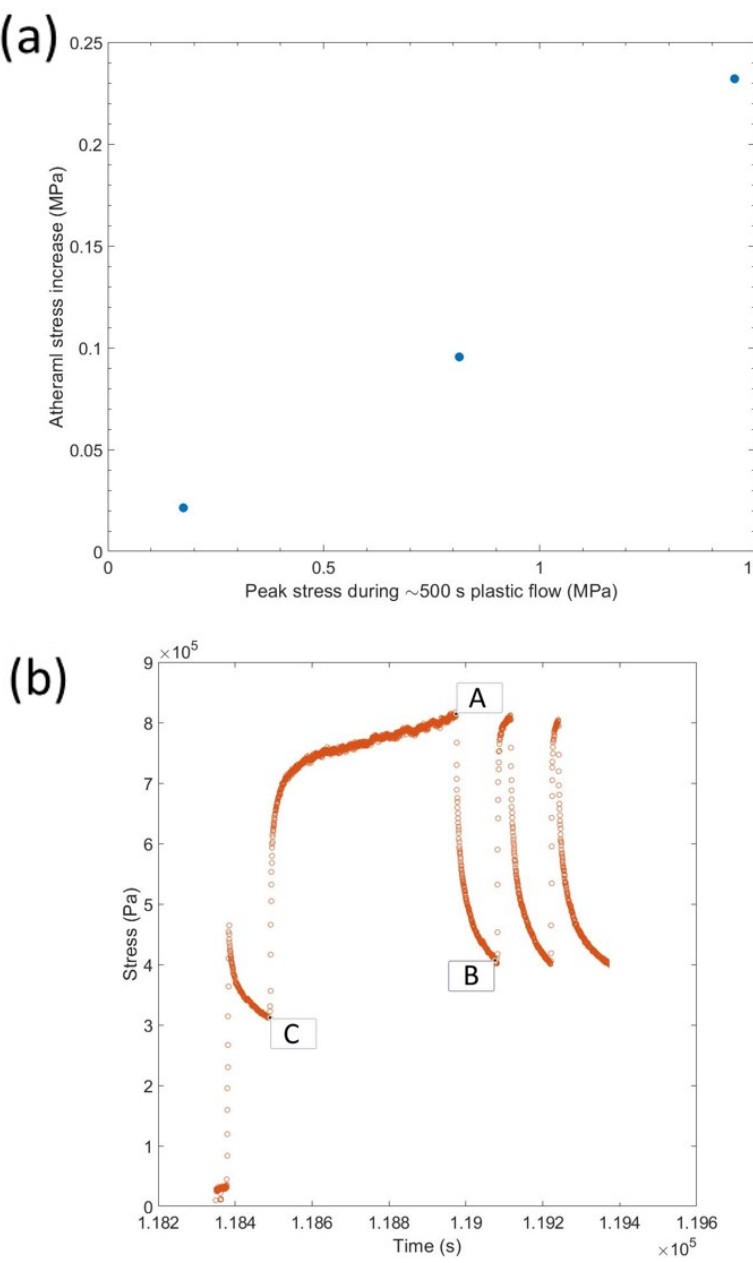

**Figure A4.** (**a**) Increase in athermal stress during plastic loading vs. the peak stress during loading for ∼500 s. (**b**) Example data showing which points are used for finding the athermal stress increase (B–C). The geometrically corrected stress after 100 s of stress relaxation at point B in (**b**) compared with the stress prior to the extended active compression at point C in (**b**), with A as the peak load.

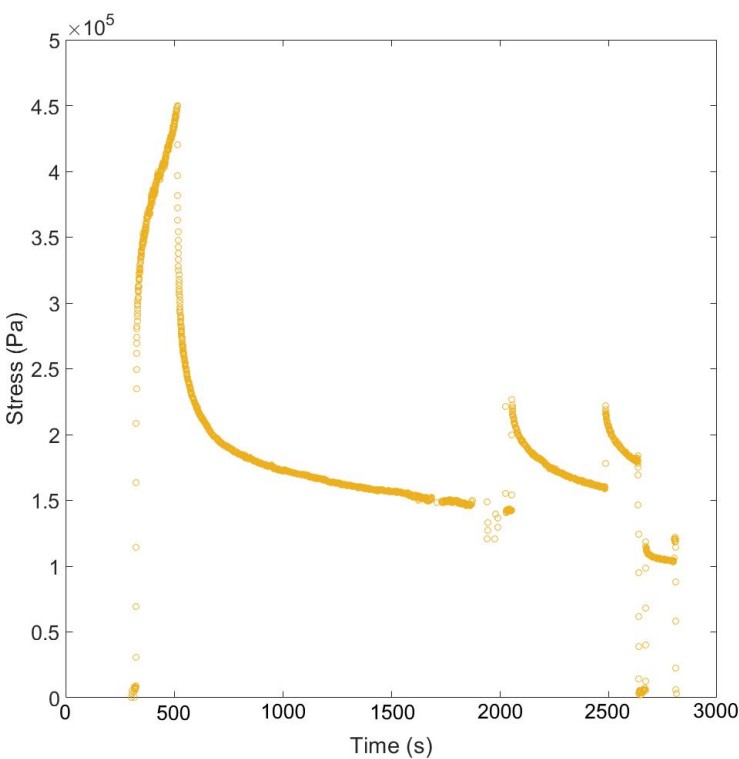

**Figure A5.** Example data showing the geometrically corrected stress over extended relaxation times (≥1000 s).

*Appendix A.5. Further Finite Element Analysis Outputs*

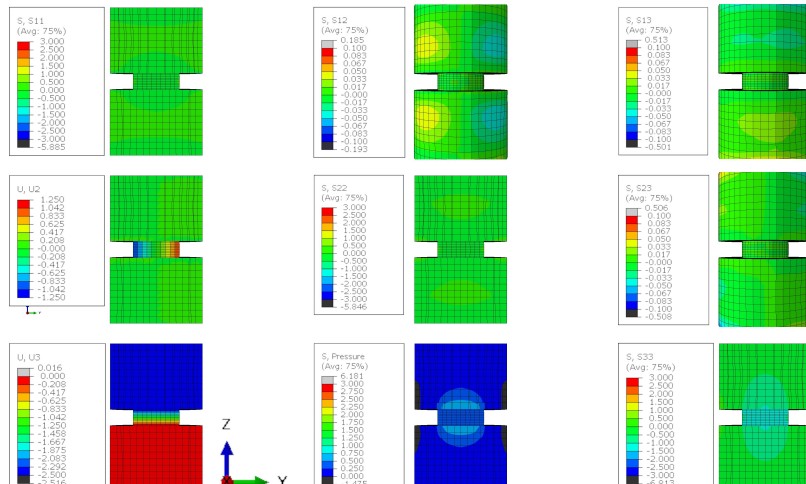

**Figure A6.** The stress state of a lithium sample compressed by 50% when the friction coefficient between the sample and the plattens is 0.01. Stress contour plots (S11, S22, S33, S12, S13, S23, and pressure) are all plotted in MPa and the displacement plots (U2 and U3) are in mm.

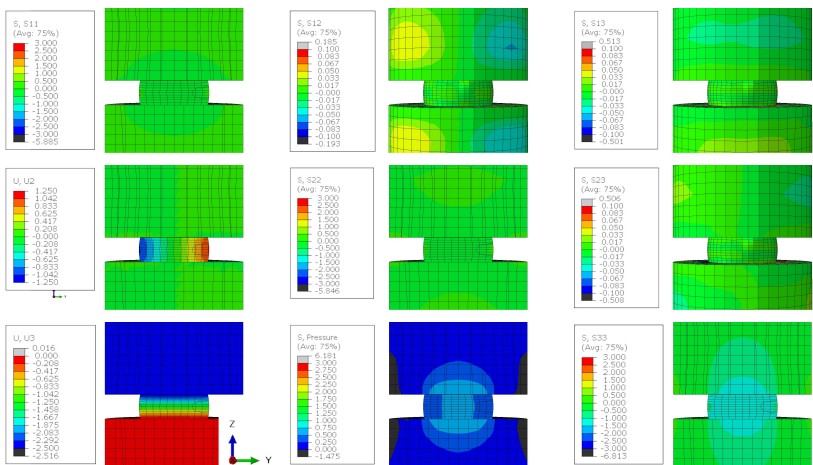

**Figure A7.** The stress state of a lithium sample compressed by 50% when the friction coefficient between the sample and the plattens is 0.1. Stress contour plots (S11, S22, S33, S12, S13, S23, and pressure) are all plotted in MPa and the displacement plots (U2 and U3) are in mm.

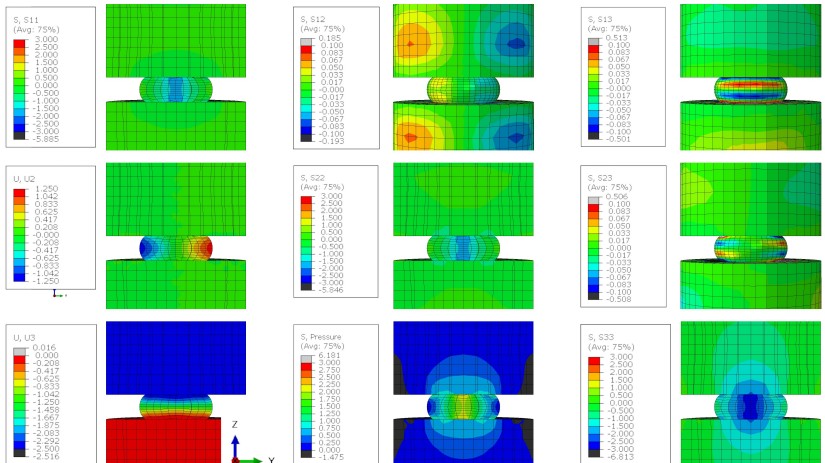

**Figure A8.** The stress state of a lithium sample compressed by 50% when the friction coefficient between the sample and the plattens is 0.5. Stress contour plots (S11, S22, S33, S12, S13, S23, and pressure) are all plotted in MPa and the displacement plots (U2 and U3) are in mm.

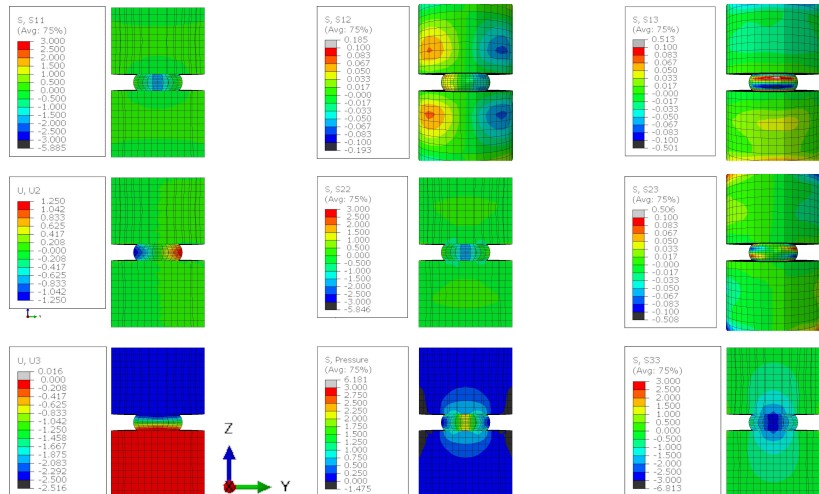

**Figure A9.** The stress state of a lithium sample compressed by 50% when the friction coefficient between the sample and the plattens is 0.9. Stress contour plots (S11, S22, S33, S12, S13, S23, and pressure) are all plotted in MPa and the dsplacement plots (U2 and U3) are in mm.

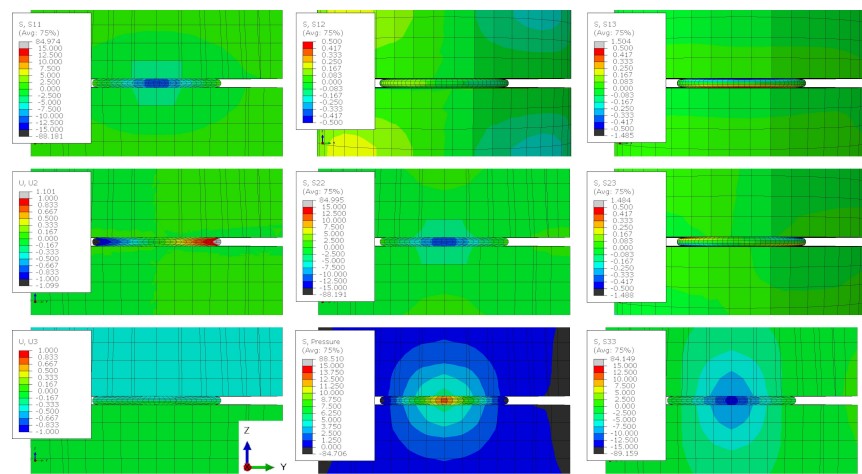

**Figure A10.** The stress state of a lithium sample compressed by 50% when the friction coefficient between the sample and the plattens is 0.5 and the height-to-width ratio is 1:5. Stress contour plots (S11, S22, S33, S12, S13, S23, and pressure) are all plotted in MPa and the displacement plots (U2 and U3) are in mm.

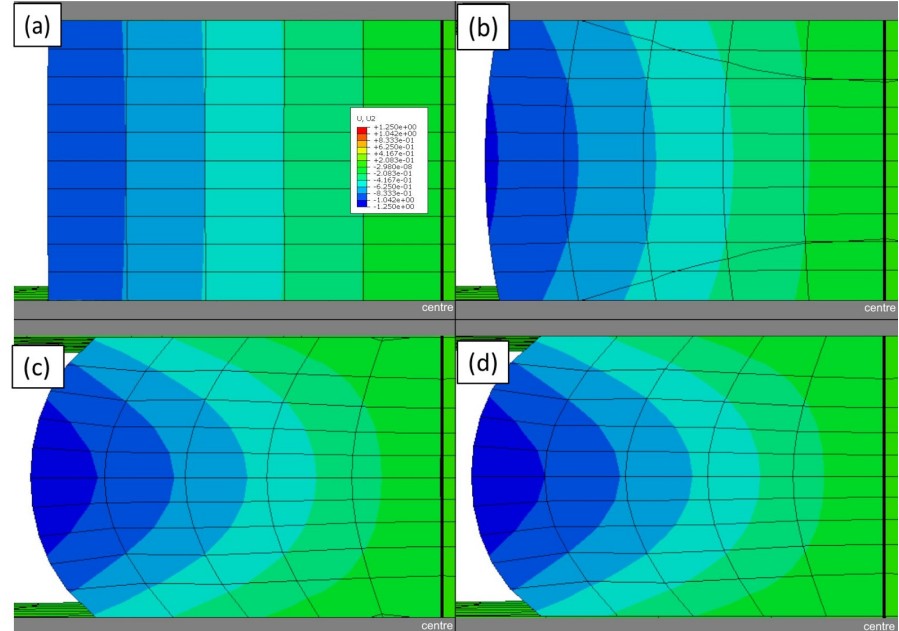

**Figure A11.** The displacement of a lithium sample compressed by 50% when the friction coefficient between the sample and the plattens is: (**a**) 0.01, (**b**) 0.1, (**c**) 0.5, and (**d**) 0.9. The displacement scale shown is in mm.

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
