# Peer review of "Lithium Metal under Static and Dynamic Mechanical Loading"

_batteries, doi:10.3390/batteries10010020_

Round 1
Reviewer 1 Report
Comments and Suggestions for Authors
This work reported the behavior of lithium metal in compression through both experimental characterizations and finite element analysis. The findings are useful and will explain the behavior of thin anode under stack pressure and during formation. Overall, the results are useful, and the manuscript is organized well. The manuscript can be accepted after the following concerns are addressed.
1. Compressive engineering stress and strain are calculated without taking any change in sample cross-sectional dimensions into account. Will this influence the results under actual working conditions?
2. How to understand the relationship between cross-section changes and hardening in the process of stress relief, and whether there is a dominant or sequential relationship?
3. What does the orange bar stand for?
Author Response
Please see the attachment
Reviewer 1
This work reported the behavior of lithium metal in compression through both experimental characterizations and finite element analysis. The findings are useful and will explain the behavior of thin anode under stack pressure and during formation. Overall, the results are useful, and the manuscript is organized well. The manuscript can be accepted after the following concerns are addressed.
- Compressive engineering stress and strain are calculated without taking any change in sample cross-sectional dimensions into account. Will this influence the results under actual working conditions?
Under the working conditions of a battery, it is unlikely that the change in anode dimensions will be measurable (without taking the cell apart and putting it back together). This is why we considered it important to report the engineering data to illustrate the response that could be measured under working conditions and then discuss how those engineering stresses and strains relate to the true changes in the material. This can be seen in the second paragraph of Results: In-Situ Observation of Compression in the original submission.
The significant increase in width under compression leads to a large increase in the contact area that in turn reduces the effective stress applied (by the same load) to the anode that will change the creep behaviour that is important for reducing void formation and growth.
- How to understand the relationship between cross-section changes and hardening in the process of stress relief, and whether there is a dominant or sequential relationship?
These changes can be considered sequentially; first under dynamic compression there will be a plastic and irreversible change in the geometry of the sample, secondly when holding that sample under a static load elastic energy stored in the material causes dislocation movement that leads to interaction and pinning. These pinned dislocations then are the source of the increased “hardening in the process of stress relief”.
- What does the orange bar stand for?
We believe that the reviewer here is referring to figure 5c). Here the orange bar is the overlapping of the red and yellow bars. To make this clearer the orange regions have been changed for a yellow and red striped pattern and an additional line of text in the figure caption for clarity.
“In the histograms b) and c), regions where bars overlap are denoted with a striped fill.”

Reviewer 2 Report
Comments and Suggestions for Authors
The presented manuscript investigates the mechanics of Lithium metal under compressive stress using experimental and computational methods. The topic is of interest for all Lithium metal containing batteries with regard to production of thin anodes and especially of interest for solid state batteries which are operated under pressure.
The authors very briefly introduce the topic and give an general overview of their work carried out. However, from the description of the experiments, it is hard to follow the results, since the authors do not state what experiments they carry out in detail, e.g. velocities and times are missing. Furthermore, there is no mention of repetition of experiments? I suggest the authors structure their description of experiments the same way they structure their results.
Almost all graphics suffer from very low axis titles and numbering making it impossible to see anything. Also, the style of graphics changes (sometimes even within one figure (e.g. Figure 1). In Figure 4 c and d it is not clear that there are 12 hours between the graphs since the x-axis is label as time. Furthermore, there are labels missing. The legends in the FME images are unreadable as well. I strongly recommend to change all the graphics with respect to readability and uniformity.
Those two point currently make it impossible to me to further judge the manuscript, hence, I recommend to consider the manuscript again after major revisions.
Author Response
Please see the attachment
Reviewer 2
The presented manuscript investigates the mechanics of Lithium metal under compressive stress using experimental and computational methods. The topic is of interest for all Lithium metal containing batteries with regard to production of thin anodes and especially of interest for solid state batteries which are operated under pressure.
The authors very briefly introduce the topic and give an general overview of their work carried out. However, from the description of the experiments, it is hard to follow the results, since the authors do not state what experiments they carry out in detail, e.g. velocities and times are missing. Furthermore, there is no mention of repetition of experiments? I suggest the authors structure their description of experiments the same way they structure their results.
The methods section has now been restructured to match the headings used in the results section with specific mention of the displacement rates (velocities) and repeats.
Almost all graphics suffer from very low axis titles and numbering making it impossible to see anything. Also, the style of graphics changes (sometimes even within one figure (e.g. Figure 1). In Figure 4 c and d it is not clear that there are 12 hours between the graphs since the x-axis is label as time. Furthermore, there are labels missing. The legends in the FME images are unreadable as well. I strongly recommend to change all the graphics with respect to readability and uniformity.
The axis numbers and titles in all figures have been changed to increase their size and improve uniformity. In Figure 4 c and d a legend has been added and an axis break has been employed to make clear the difference in time between data measurements. New larger legends have been added to the finite element analysis outputs (Appendix A.5)
Those two point currently make it impossible to me to further judge the manuscript, hence, I recommend to consider the manuscript again after major revisions.
For ease I have included all the updated figures here and the reorganized text of the methods below:
